# The Energy and Nutritional Value of Meat of Broiler Chickens Fed with Various Addition of Wheat Germ Expeller

**DOI:** 10.3390/ani13030499

**Published:** 2023-01-31

**Authors:** Zuzanna Goluch, Maja Słupczyńska, Andrzej Okruszek, Gabriela Haraf, Monika Wereńska, Janina Wołoszyn

**Affiliations:** 1Department of Food Technology and Nutrition, Wroclaw University of Economics and Business, 53-345 Wroclaw, Poland; 2Department of Animal Nutrition and Feed Science, Wroclaw University of Environmental and Life Sciences, 51-631 Wroclaw, Poland

**Keywords:** chicken, meat, wheat germ expeller, energy value, nutritional value, nutrient reference values

## Abstract

**Simple Summary:**

The study concerns the effect of wheat middling’s replacement with 5, 10, and 15% wheat germ expeller, as a feed additive given to broiler chickens, on their meat’s metabolizable energy and nutritional value. The breast and thigh muscles of chickens were analyzed. The additive wheat germ expeller did not affect the muscles’ energy values but did affect the nutritional value. The daily consumption of 100 g of breast muscles from broilers to a large extent covers the consumer nutrient reference values-requirements (NRV-R) for P, Mg Fe, Cu, and Mn. However, thigh muscles cover the NRV-R to a greater extent for Ca and Zn. Adding 5% WGE to broiler feed is optimal as it does not impair the nutritional value of the muscles.

**Abstract:**

The study concerns the effect of wheat germ expeller (WGE) as a feed additive given to male Ross-308 broiler chickens on their meat’s energy and nutritional value, and coverage of nutrient reference values-requirements (NRV-R) of consumers for particular minerals. The chickens in the control group (CT—Control Treatment) were fed a standard complete mix. The experimental groups (EX5, EX10, EX15) were given a feed in which wheat middling was replaced with 5, 10, and 15% WGE. The breast and thigh muscles of 32 randomly selected chickens (8 in each group) were analyzed. More water, crude protein, P, Mg, Fe, Cu, and Mn were determined in the breast muscles, and more crude fat, crude ash, Ca, and Zn in the thigh muscles. Chickens from the CT group consumed significantly (*p* ≤ 0.01) less feed per body weight than those from groups EX5 to EX15, but achieved the highest body weight per 100 g of consumed feed. A higher (*p* ≤ 0.01) feed, energy, crude protein, and crude fat intake was observed in groups EX5 to EX15 compared to CT. The higher (*p* ≤ 0.01) value of protein efficiency ratios was indicated in the CT group. The WGE additive did not impact the muscles’ energy values but affected the nutritional value. The daily consumption of 100 g of breast muscles to a large extent covers the consumer NRV-R for P, Mg Fe, Cu, and Mn. However, thigh muscles cover the NRV-R to a greater extent for Ca and Zn. The EX15, EX5, and EX10 muscles covered most of the NRV-R for P, Ca, and Mg, while the CT muscles did the same for Zn and Mn. Adding 5% WGE to broiler feed is optimal as it does not impair the nutritional value of the muscles.

## 1. Introduction

Poultry meat is regarded by the UN Food and Agricultural Organization [1] as a commonly available and inexpensive food that is rich in nutrients. Therefore, it is produced in many countries of the world.

The United States is the world’s largest chicken meat producer, with 40.5% of global output, followed by China and Brazil, in 2020 [2]. Chickens contribute 90% of global poultry meat production, followed by turkeys (5%), ducks (4%), and geese and guinea fowl (2%). According to the forecasts of the Organization for Economic Co-operation and Development (OECD) [3] for the years 2021–2030, there will be an increase in consumer demand for meat (as long as it is healthy and of high quality), despite plant-based diets and meat substitutes being promoted in the media. There is a clear trend of poultry meat consumption being on the rise in virtually all countries and regions. Consumers are attracted to poultry due to lower prices, product consistency and adaptability, and higher crude protein/lower crude fat content. Consumption of poultry meat is projected to increase globally to 152 Mt (metric tons) over the projection period, accounting for 52% of meat consumption. Poultry meat production will expand rapidly due to an increase in sustained productivity in China, Brazil, and the United States, and investments made in the European Union (due to lower production costs in Hungary, Poland, and Romania). In 2021, Poland exported 1.8 million tons of poultry products, which is 0.4% more than in 2020 [4]. Contemporary consumers consider poultry’s place of origin and prefer poultry from local or regional farms that promote the well-being of the birds [5]. However, a key factor that encourages consumers to include poultry meat in their diets is its energy, nutritional values, and health properties [6,7], which are influenced mainly by how chickens are fed [6].

Complete mix feeds for broilers contain soya middlings. Still, the tendency is to use local or regional ingredients or even, according to the Farm to Fork Strategy of the European Union, by-products of the food industry [7,8,9]. The EU Commission Regulation [10] allows using wheat germ expeller in the feed, which is a product of oil manufacture obtained by pressing wheat germ (*Triticum aestivum* L., *Triticum durum* Desf.) and other cultivars of wheat and dehusked spelt (*Triticum spelta* L., *Triticum dicoccum* Schrank, *Triticum monococcum* L.), to which parts of the endosperm and teste may still adhere (item 1.11.13 in the UE Commission Regulation). Using these kinds of by-products in animal feeds fits into a current trend of production based on natural components. Simultaneously, consumers are willing to pay higher prices for poultry products perceived as natural or environmentally friendly and produced on farms with high animal welfare and animal nutrition standards [11]. Source literature lacks research concerning the nutritional value of the meat of slaughter animals fed with an additive of wheat germ expeller.

While the liver plays the main physiological role in maintaining the homeostasis of trace elements, they are distributed from the liver to other organs and tissues, including muscles, where they are deposited. The skeletal muscles are one of the organs most adaptable to environmental changes, which include, among others, nutrition. Chicken muscles, as culinary meat, are more often consumed by humans than the liver, an edible by-product [12]. It should also be noted that consumers use different types of culinary treatment of meat, which will also affect its nutritional value. Depending on the type of thermal treatment of meat, minerals might be lost, and mineral components, which can be found in the form of soluble dissociated salts (part of sodium, small amounts of phosphorus, calcium, and potassium), end up in the leakage. Components that combine with proteins, such as iron, remain in the meat [13]. Therefore, from the potential consumer’s point of view, information on the energy and nutritional value of raw meat obtained from broiler chickens fed with an alternative feed additive may influence their purchasing decisions. Therefore, the study aimed to (1) determine the energy and nutritional value of Ross-308 chicken breast muscles and thighs after adding to the feed (by replacing wheatmeal) 5%, 10%, and 15% wheat germ expeller, (2) calculate what percentage of 100 g of this meat covers the nutrient reference values-requirements of the consumer for individual minerals.

## 2. Materials and Methods

### 2.1. Ethics Statement 

The Advisory Team approved the experiment for the Welfare of Animals in The Faculty of Biology and Animal Science of Wroclaw University of Environmental and Life Sciences (Decision no. 1/2019). The study protocol did not require the approval of the Ethics Committee. The chickens were maintained according to European Union and Ethical Commission regulations [14].

### 2.2. Experimental Design and Diets

The dietary experiment (a pilot study) was conducted on 112 Ross-308 broiler chickens from the Złotoryja Hatchery in Poland (PL 16096409, Ferma Tarnica 1, 41-156 Gracze, Poland). Male one-day-old chicks were vaccinated with Poulvac IB Primer (Zoetis Polska Sp. z o.o., Warszawa, Poland) against infectious bronchitis and with Nobilis ND C2 (Intervet International B.V., AN Boxmeer, The Netherlands) against Newcastle disease virus, and randomly allocated into 28 battery cages (4 birds per cage), which constituted 4 groups (7 replications per group). The maximum bird density was lower than 16 kg/m^2^, following the requirements of European Council Directive 2007/43/EC [15]. During the first day of the experiment, the temperature in the room was 32–33 °C and was systematically lowered the following days, to 20–22 °C on the last day. Moreover, an intermittent light program was used, with a repeated light and dark cycle of 18 and 6 h, respectively (with the exclusion of the first 2 days of the grow-out, when the chicks had access to light 24 h according to the requirements included in the information brochure concerning Ross-308 chicken grow-out [16]).

During the experiment, the chickens had constant access to water. Chickens from the control group (CT—Control Treatment) were fed standard, isoenergetic and isoprotein mixes ad libitum: starter—from day 1 to day 10, grower—from day 11 to day 25, and finisher—from day 26 to day 43 of life; the feeds were based on wheat, corn, and post-extraction soya middlings. All diets were served in a loose form, and their formulation was adjusted to the feeding requirements of Ross-308 chickens [17]. On the other hand, chickens from all three experimental groups, EX5, EX10, and EX15, were given feed where wheat middlings were replaced with 5, 10, and 15% of wheat germ expeller, respectively. The dietary ingredients used in the experiment are presented in Table 1. Metabolic energy was calculated for feed components based on the results of chemical analyses before formulating diet recipes, per the guidelines of the Polish Nutritional Recommendations and Nutritional Value of Feeds for Poultry [18]. The chemical composition of the wheat germ expeller and experimental diets is shown in Table 2 and Table 3.

The body weights of the birds were measured on day 1 and on days when the complete feed mix was switched into the grower feed (day 11), the finisher feed (day 26), and the last day of the experiment (day 43). It was measured using an electronic scale made by Radwag (Radom, Poland) to the nearest 0.1 g.

Twelve hours before the end of the experiment, the chickens were taken off the feed and provided only with water. On the last day of the experiment, 8 birds from each group were randomly selected and slaughtered, according to the procedures specified in Annex IV of Directive 2010/63/EU [19] of the European Parliament and Council [7]. 

After slaughtering the birds, the breast and thigh meat were excised by removing the skin, bones, and connective tissue. The breast and thigh muscles from the left and right sides of the carcasses were dissected after chilling at 4 °C for 24 h. Then, they were trimmed of visible skin, excess fat, and connective tissues. The breast meat samples were ground separately, using a meat grinder. The samples were subsequently divided into aliquots for meat composition analysis. Finally, the samples were poured into plastic sample bottles and were kept in a freezer at −18 °C for further analysis.

### 2.3. Chemical Composition of Feedstuffs and Meat

The basic chemical content of feedstuffs and breast and thigh meat were analyzed using reference methods, following the official analytical methods of EN ISO 9831:2004 [20] and the Association of Official Analysis Chemists (AOAC) [21]. Sample preparation, proceedings, and the equipment used during the analyses followed the procedure given by Goluch et al. [13].

### 2.4. Mineral Analysis

The minced meat was frozen for 12 h at −18 °C, and freeze-dried for about 48–72 h (under pressure, −55 °C), depending on the size of the sample, in the freeze-drying apparatus from Edwards Modulyo (Akribis Scientific Limited, Knutsford, UK) to achieve a constant mass of the sample. Next, the freeze-dried samples were ground in a laboratory grinder, WŻ-1 (Zakład Badawczy Przemysłu Piekarskiego Sp. z o.o., Bydgoszcz, Poland).

The samples (0.3 g for Ca, Mg, K, Na, and 1.0 g for Cu, Mn, Zn, Fe) of the freeze-dried broiler meat were wet digested with 7 mL of HNO_3_-H_2_O_2_ mixture (2:3, *v*/*v*) using a MarsXpress microwave oven (MARS 6 Microwave Reaction System, CEM Corporation, Matthews, NC, USA). The digestion program was the following: first (10 min), the temperature was increased to 190 °C; second (7 min), the temperature was kept at 195 °C. Digested samples were placed in polypropylene tubes and diluted to 50 mL with ultra-pure water. A blank digest was made in the same way. The concentrations of macro- (K, Na, Ca, Mg) and microelements (Zn, Fe, Mn, Cu) were determined, using a flame atomic absorption spectrometry (FAAS, air-acetylene flame), with a AA 240FS SIPS20 spectrometer (Varian, Mulgrave, Australia), according to the AOAC procedures [22]. The content of macro- and microelements in the samples was expressed in milligrams per 100 g dry mass (DM).

The P content in broiler meat was evaluated after previous mineralization samples with HNO_3_ (65%) and HClO_4_ acid in close microwave mineralizer MarsX9 (MARS 6 Microwave Reaction System, CEM Corporation, Matthews, NC, USA). It was analyzed spectrophotometrically by the ammonium vanadomolybdate method, using a Specol 11 (Carl Zeiss, Jena, Germany) at a wavelength of 470 nm [22]. The content of P was expressed in milligrams per 100 g DM.

Wheat flour was used as the standard reference material SRM 1567b^®^ to determine the mineral composition of feedstuffs and meat samples (National Institute of Standards and Technology, USA, https://www.nist.gov/srm (accessed on 3 December 2022). The determined concentrations (mg × kg^−1^) of Ca, P, Mg, K, and Na were 198 ± 20.4, 1198 ± 127, 356 ± 5.3, 1304 ± 158, and 6.74 ± 0.80 (*n* = 3), respectively, with all macroelements recovery ranging from 89.4% to 103.4%. Fe, Zn, Mn, and Cu concentrations were also tested using SRM 1567b^®^ with trace minerals recovery, ranging from 97.0 to 104.2%.

### 2.5. Indices

The feed conversion ratio (FCR) factor was calculated with the following formula: FCR% = total feed intake (g)/final body weight (g)(1)

The protein efficiency ratio (PER) was calculated with the following formula [23]:PER = weight gain (g)/weight of protein consumed (g)(2)

### 2.6. Statistical Analysis 

The results were verified for normality distribution with the Shapiro–Wilk Test and homogeneity variation with Laven’s test. The findings were log-transformed to attain or approach a normal distribution, and, subsequently, a two-way analysis of variance (ANOVA) was made. Statistically significant differences between the averages of the groups were calculated using Tukey’s multiple-comparison test, on the level of significance *p* ≤ 0.05 and *p* ≤ 0.01, with the use of Statistica^®^ 13.1 software [24]. The tables show arithmetic means and standard errors of the means (SEM). All data are reported as means of 2 parallel measurements.

## 3. Results and Discussion

### 3.1. Growth Performance

The effects of the different dietary treatments on the growth performance of broiler chickens are presented in Table 4.

The results of this study show differences in the daily feed intake (calculated per 100 g of body weight of broiler chickens) between the CT and the experimental groups. Chickens from the CT group consumed significantly (*p* ≤ 0.01) less feed per body weight than those from groups EX5 to EX15 However, the highest increase in body mass per 100 g of consumed feed was achieved by the chickens from the CT group, compared to the experimental groups EX5 to EX15.

Using 5% and 10% wheat germ expeller as a feed additive significantly impacted the increase in gross energy, protein, and fat consumption by the broilers in the experimental groups, compared to the CT group. The highest (*p* < 0.01) consumption of the feed and its worst use were noted in the EX10 group, which may result from a higher crude protein content to the gross energy content in the feed (Table 3). However, the chickens from group CT, in comparison to the experimental groups, were characterized as having significantly (*p* < 0.01) the lowest consumption of feed to the final body weight and consumed protein per achieved body weight. A slightly higher value of the indicator FCR (1.88) for Ross-308 chickens after 42 days of feeding was established by Shakouri and Malekzadeh [25] and Gornowicz et al. [26] as it was 1.88 and 1.86 kg/kg, respectively.

The reason for a smaller feed intake in the CT group of broilers might be a smaller consumption of P, Na, Ca, Mg, Fe, Zn, and Mn compared to the experimental groups (Table 4). The highest intake of P, Na, Ca, and Zn per 100 g of body weight was found in group EX10, while K and Mg were highest in group EX5, and Fe and Mn were highest in group EX15. This resulted from the content of the abovementioned mineral elements in given feeds (Table 3) and their consumption (Table 4). However, no differences were found in the intake of Cu in various groups of broilers.

It should be stressed, however, that a higher content of mineral elements in the feed that comes from natural feed ingredients, and not from synthetic supplements, is a positive phenomenon because it does not demand a higher metabolism of xenobiotics in the livers of chickens, nor does it require a better excretion of its metabolites through kidneys or with bile [27]. Mineral compounds of natural origin also reduce environmental pollution by limiting their excretion with chickens’ excrements [28]. Moreover, mineral ingredients deposited in the edible chicken tissue do not pose any health risks to the consumer.

### 3.2. Proximate Composition 

Contemporary consumers pay attention to the energy value of food; they are also aware of the impact of quantity, quality, and health properties of food on the human body [29]. Broiler chicken meat is perceived by consumers as healthier than red meat due to its high protein and lower fat and cholesterol content. When buying poultry meat, consumers take into account its availability, low price, high palatability, and nutritional value [30].

The determined gross energy value and basic chemical composition of the breast and thigh muscles of Ross-308 chickens are presented in Table 5.

In terms of the type of muscles, irrespective of the experimental group, it was indicated that the thigh muscles had a significantly (*p* < 0.01) higher gross energy value, crude fat content, and crude ash than the breast muscles. In contrast, it was demonstrated that there is a significantly (*p* < 0.01) higher water and crude protein content in the breast muscles than in the thigh muscles. Differences in the content of the components mentioned above result from the structure and metabolic functions of the muscles [31]. The breast muscle has more high-activity glycolytic fibers than the thigh muscles. It is generally characterized by a lower intramuscular fat content and a greater water and protein content [32,33]. A smaller (0.46 MJ/ 100 g Wet Mass) energy value of the breast muscles and a similar energy value of the thigh muscles (0.95 MJ/ 100 g WM) in Ross-308 chickens after 42 days of rearing (but calculated based on converters for fat and protein) were found by Haščík et al. [34].

The energy value of the muscles of the tested broilers did not differ significantly between the groups, despite the higher energy intake in the feed with the addition of a germ expeller (Table 4).

There was no significant interaction between the type of muscle and the group of tested chickens in terms of energy value and water and fat content in the conducted study. 

There were no significant differences between groups of broilers in energy value and the content of water, crude protein, crude fat, and crude ash in the breast muscles. The determined content of the ingredients mentioned above in the CT group, calculated in the dry matter (DM), was similar to those found by Proskin et al. [35] in the breast muscles of Ross-308 chickens: (water 77.67%, protein 20.32% DM), but lower fat (1.85% DM) and ash (1.28% DM) content were found after 42 days of their rearing [35]. However, in the current research, contents of water, crude protein, crude fat, and crude ash in the breast muscles of the CT group, calculated as wet mass (WM), were, respectively: 73.5%; 22.9%; 2.40%; and 1.03%. These results were similar to findings reported by Martínez and Valdivié [36] concerning Ross-308 chickens after 35 days of rearing (73.3%; 23.9%; 1.15%; and 1:7% WM, respectively). Similarly, other authors [37,38,39] obtained similar results in the breast muscles of the same breed of chickens after 42 days of rearing: water (72.65%; 74.0%; and 73.9%), protein (22.88%; 22.4%; and 23.03%WM), fat (2.03%; 2.49; and 1.38% WM), and ash (2.44%; 1.13; and 1.17% WM).

The addition of germ expeller significantly affected the content of water, crude protein, and crude ash in broiler thigh muscles. The highest (*p* ≤ 0.01) moisture and crude protein contents were found in the CT group thigh muscles compared to EX10 and EX15. In relation to crude ash, the highest content (*p* < 0.05) was found in the thigh muscles of chickens in EX10 compared to the CT group. Gornowicz et al. [37] and Biesiada-Drzazga et al. [39] found similar moisture (72.0% and 74.69% WM), protein (18.63% and 19.11% WM), and ash (1.53% and 1.0% WM) content in the thigh muscles of Ross-308 chickens from the control group after 42 days of rearing. In our study, the crude fat content determined in the thigh muscles in the CT group was similar to that found by Gornowicz et al. [37], but higher than that found by other authors [34,38,39], respectively: 13.2%, 7.69%, and 5.52% WM.

A significant (*p* < 0.01) interaction between the type of muscle and the groups of chickens studied was found only in the crude protein and crude ash content of the muscles.

### 3.3. Macroelements

P, Na, Ca, K, and Mg are significant among the minerals of poultry muscles. The main task of Mg, Na, and K is maintaining homeostasis between the cations Ca^2+^, K^+^, Mg^2+^, and the anions Cl^−^, HCO_3_^−^, and PO_4_^3−^. 

The macroelements composition of the breast and thigh of the Ross-308 broiler is shown in Table 6.

The highest content (*p* ≤ 0.01) of P, Na, K, and Mg were found in the breast muscles compared to the thigh muscles. A significantly (*p* ≤ 0.01) higher Ca content was found in the thigh muscles than in the breast muscles. Straková et al. [40] found in the breast and thigh muscles of 40-day-old roosters Ross-308 the P, Ca, and Mg content greater than our study (951; 215; and 154 vs. 760; 180; and 112 mg/g DM, respectively).

However, in our studies, the determined contents of P, Na, Ca, K, and Mg in the breast and thigh muscles of the CT group in terms of fresh weight (WM) were respectively: 240.3; 102.5; 5.91; 197.5; and 29.2 vs. 221.8; 92.0; 11.10; 135.7; and 25.7 mg/100 g. These findings partly coincide with the USDA database data [41], according to which the raw breast muscles of skinless broiler chickens contain a greater amount of P (213 mg/100 g WM), K (334 mg/100 g WM), and Mg (28 mg/100 g WM) than the thigh muscles (analogically: 185 mg; 242 mg; and 23 mg/100 g WM). On the other hand, according to the cited database, similarly to our research, breast muscles contain less Na than thigh muscles (45 mg vs. 95 mg/100 g WM). Yet, these data do not refer to broilers’ specific breed and sex. However, in the same breed of chickens, Al-Yasiry et al. [42] found a higher content of P, Ca, and Mg in the breast muscles than in the thigh muscles in the control group (24.0 mg; 2.80 mg; 1.64 mg vs. 19.5 mg; 0.81 mg; 0.21 mg/100 g WM). In the thigh muscles, however, they determined a higher content of Fe, Zn, and Cu than in the breast muscles (0.065 mg; 0.15 mg; and 0.008 mg vs. 0.047 mg; 0.050 mg; and 0.008 mg/100 g WM). In the studies conducted on 42-day-old Ross-308 roosters, Suchý et al. [38] showed a higher content of P (225 mg vs. 204 mg/100 g WM), less Mg (11 mg vs. 13 mg/100 g WM), and the same content of Ca (30 mg/100 g WM) in the breast muscles compared to the thigh muscles.

Feed additives of 5% (EX5) and 10% (EX10) of germ expeller significantly (*p* < 0.05) contributed to a total increase in the phosphorus content in the muscles, compared to the CT group. This is the result not only of a higher content of this compound in the feed (Table 3), but also of its higher consumption (*p* ≤ 0.01) by broilers from these groups compared to the CT group (Table 4). Although Ca:P (not cited in the paper) in the feed from the EX5 and EX10 groups was higher than in CT, it was similar in all groups’ broilers’ muscles.

A significant interaction between the muscle type and the studied chickens’ experimental groups was found for P, Na, and Ca. The use of the germ expeller feed additive significantly increased the breast muscle content of Na (EX5–EX15), Ca (EX5 and EX10), and Mg (EX10 and EX15) compared to the CT group. On the other hand, in the thigh muscles, there was a significantly higher content of P (EX15), K (EX5–EX10), and less Na (EX10 and EX15) compared to the CT group (Table 6).

### 3.4. Microelements

Among the microelements studied, Fe, Cu, Zn, and Mg belong, on the one hand, to the essential components in the body of humans and animals, and, on the other hand, to heavy metals [43]. Therefore, their content in the muscles of animals for slaughter and poultry should be at a physiological level so as not to create a health risk for both animals and consumers of meat obtained from them.

The microelements composition of the breast and thigh of the Ross-308 broiler is shown in Table 7.

When analyzing the type of muscles, significantly (*p* ≤ 0.01) higher contents of Fe, Cu, and Mn in the breast muscles were found compared to the thigh muscles. Kokoszyński et al. [44] showed that in 6-week-old Roosters Ross-308, there was a higher K, P, and Mg content in the breast muscles than in the thigh muscles (162.0; 176.0; and 160 vs. 114.0; 148.0; and 97.5 mg/100 g DM). In contrast, in the thigh muscles, the content of Na, Zn, and Fe (294.0; 49.5; and 55.5 vs. 192.1; 28.3; and 14.9 mg/100 g DM) was lower than in the breast muscles [44,45].

However, in our studies, the contents of Fe, Zn, Cu, and Mn in the breast and thigh muscles of the CT group, calculated in terms of fresh weight (WM), were respectively: 2.30; 0.71; 0.30; and 0.043 vs. 2.06; 1.36; 0.21; and 0.038 mg/100 g. These results are different from those contained in the USDA database [41], where the raw breast muscles of skinless broiler chickens have a smaller amount of Fe (0.37 mg/100 g WM), Cu (0.037 mg/100 g WM), and Mn (0.011 mg/100 g WM) than the thigh muscles (analogically: 0.81 mg; 0.062 mg; and 0.013 mg/100 g WM). However, in our research, we found a significantly (*p* ≤ 0.01) higher content of Zn in the thigh muscles (1.36 mg/100 g WM) than in the breast muscles (0.71 mg/100 g WM), which is consistent with the data in the USDA database (1.58 vs. 0.68 mg/100 g WM).

The use of germ expeller as a feed additive caused a significant change in the muscles’ contents of Zn and Mn. The highest Fe content was found in the EX15 group compared to EX10. In the breast muscles, the significantly (*p* < 0.05) highest content of Zn in the CT group was found, compared to EX10. On the other hand, in this group of broilers, the significantly lowest content of Zn in the thigh muscles was found compared to the other groups. The Mn content was the highest (*p* ≤ 0.01) in CT muscles compared to EX5 and EX10.

A significant (*p <* 0.05) interaction between the type of muscle and the group of chickens studied was found only for Fe.

The obtained contents of macro- and microelements in the chicken muscles under the influence of the germ expeller used cannot be compared with the results of other authors because no similar literature on the subject has been found.

### 3.5. Nutrient Reference Values—Requirements

For the informed consumer, it is important to indicate the energy and nutritional value of the food product, which, following the applicable regulations, should be placed on its label. Nutrient Reference Values (NRVs) are values used in nutrition labeling derived from authoritative recommendations for daily nutrient intake. Nutrition labeling provides information about the food’s key nutrient content to inform consumers about the nutritional quality of the foods they purchase. These recommendations are based on the best available scientific knowledge of the daily energy or nutrient needed for good health. NRVs do not appear on the label, but they are used in nutrition labeling to show the contribution to healthy nutrient intake of the nutrients in a portion of food [45]. However, no NRV has been established for sodium and potassium.

Nutrient Reference Values-Requirements and their implementation in the breast and thigh of a Ross-308 broiler are shown in Table 8.

In the conducted studies, it was found that the daily intake of 100 g of breast muscles of broilers, compared to the thigh muscles, in a significantly (*p* ≤ 0.01) larger percentage covers the consumer’s NRV-R for P, Mg, Fe, Cu, and Mn. In contrast, the same amount of thigh muscle consumed, compared to breast muscles, in a significantly (*p* ≤ 0.01) larger percentage covers the consumer’s NRV-R for Ca and Zn.

The demand for P, Ca, and Mg is best covered by muscles from the EX15, EX5, and EX10 groups, respectively, while for Zn and Mn, it is best covered by the CT muscles.

A significant interaction between the muscle type and the group of chickens studied in the implementation of NRV-R was found for P, Ca, and Fe. The highest NRV-R realization for P was found for 100 g of thigh muscles EX15, compared to the CT group and EX5. In the case of Ca, the highest NRV-R realization was found in the breast muscles EX5 and EX10, compared to CT and EX15. On the other hand, in the significantly highest percentage, the breast muscles of the CT group meet NRV-R for Zn, compared to EX10. Thigh muscles from the EX10 group in the lowest rate meet NRV-R for this element compared to other groups of chickens.

## 4. Conclusions

The research shows that the chicken breast muscles were characterized by lower energy values and contained more moisture, crude protein, P, Mg, Fe, Cu, and Mn. On the other hand, with a higher energy value and lower water content, thigh muscles contained more crude fat, crude ash, Ca, and Zn. The daily intake of 100 g of chicken breast muscle in a larger percentage meets the consumer’s NRV-R for P, Mg, Fe, Cu, and Mn than the muscles of the thighs. In contrast, the thigh muscles in a larger percentage meet the consumer’s NRV-R for Ca and Zn.

The use of various additives of germ expeller did not significantly affect the energy value of the muscles, but it did affect for their nutritional value, which resulted from the composition of the feed and the amount of its consumption. Considering the use of germ expeller, the demand for P, Ca, and Mg are met in the highest percentage by muscles from the EX15, EX5, and EX10 groups, respectively, while the Zn and Mn are from the CT group. Therefore, the information obtained can guide consumers of poultry meat when making their dietary choices.

It seems that a 5% addition of germ expeller would be optimal because without worsening the basic chemical composition of the muscles, it increased the content of P, Na, and Ca in them, compared to the CT group, although with slightly less effective indicators of rearing chickens. As is well known, feed costs can account for up to 70% of total production costs. Therefore, it is reasonable to continue looking for ways to optimize the consumption of germ expellers by broiler chickens while considering the functional features of the meat they obtained. It may turn out that even with higher costs of rearing chickens, the energy and nutritional value of this meat will arouse the interest of conscious consumers, producers of poultry livestock, and its processors.

## Figures and Tables

**Table 1 animals-13-00499-t001:** Dietary ingredients content, metabolisable energy (ME) value (unit: MJ), and essential nutrients of experimental diets [g/kg of feed].

Ingredients	Starter (1–10 D)				Grower (11–25 D)				Finisher (26–43 D)			
	CT ^1^	EX5 ^2^	EX10 ^3^	EX15 ^4^	CT ^1^	EX5 ^2^	EX10 ^3^	EX15 ^4^	CT ^1^	EX5 ^2^	EX10 ^3^	EX15 ^4^
Ground maize	403.0	432.9	459.9	486.7	443.7	470.6	496.5	523.4	446.5	472.4	499.3	526.2
Ground wheat	150	100	50	0	150	100	50	0	150	100	50	0
Soybean meal	358	326	296	266	304	274	245	215	303	274	244	214
Rapeseed oil	48	50	53	56	60	63	66	69	65	68	71	74
Wheat germ expeller	0	50	100	150	0	50	100	150	0	50	100	150
Sodium bicarbonate	5.40	5.40	5.32	5.32	5.77	5.74	5.72	5.69	5.77	5.75	5.72	5.68
Monocalcium phosphate	13.7	13.7	13.6	13.6	12.9	12.8	12.8	12.7	12.0	12.0	11.9	11.9
Limestone	13.7	13.8	14.0	14.2	14.1	14.3	14.5	14.7	12.6	12.9	13.1	13.3
L-Lysine	2.14	2.21	2.24	2.27	2.86	2.89	2.89	2.92	0.84	0.86	0.87	0.91
DL-Methionine	2.77	2.61	2.45	2.29	2.73	2.57	2.40	2.25	1.66	1.50	1.33	1.17
L-Threonine	0.71	0.87	1.01	1.15	1.49	1.62	1.75	1.90	0.07	0.21	0.34	0.48
Premix 0.25%	2.50	2.50	2.50	2.50	2.50	2.50	2.50	2.50	2.50	2.50	2.50	2.50
ME value and essential nutrients of experimental diets												
ME	12.50	12.49	12.50	12.50	12.98	12.99	12.99	12.99	13.19	13.19	13.19	13.19
Dry matter	892	899	905	910	893	901	907	912	893	901	906	911
Crude protein	221	220	220	220	200	200	200	200	200	200	200	200
Crude fiber	28.3	27.2	26.2	25.1	27.1	26.1	25.0	24.0	27.1	26.1	25.0	24.0
Ca	9.45	9.40	9.40	9.40	9.20	9.21	9.20	9.20	8.50	8.51	8.50	8.50
P available	4.30	4.30	4.30	4.30	4.00	4.00	4.00	4.00	3.80	3.80	3.80	3.80
Na	1.60	1.61	1.60	1.60	1.70	1.70	1.70	1.70	1.70	1.70	1.70	1.70
Lysine (total)	12.00	12.00	12.00	12.00	11.51	11.51	11.50	11.50	9.50	9.52	9.50	9.51
Methionine (total)	5.50	5.50	5.50	5.50	5.20	5.20	5.20	5.21	4.30	4.30	4.30	4.30
Threonine (total)	8.01	8.00	8.00	8.00	8.01	8.00	8.00	8.01	6.60	6.61	6.60	6.60

^1^ CT, Control Treatment; ^2^ EX5, Experimental diet with 5% wheat germ expeller; ^3^ EX10, Experimental diet with 10% wheat germ expeller; ^4^ EX15, Experimental diet with 15% wheat germ expeller. The content of additives added in the premix (0.25%) in 1 kg of the mixture: Vitamin A—8100 UI; D3—3000 UI; E—22.50 UI; K3—1.25 mg; B1—1.25 mg; B2—4.60 mg; B6—2.20 mg; B12—0.02 mg; PP—20.0 mg; choline chloride—184.25 mg; calcium D-pantothenate—6.50 mg; folic acid—0.34 mg; biotin—0.10 mg; betaine; hydrochloride—41.89 mg; Cu—10 mg; Fe—30 mg; Zn—60 mg; Mn—70.40 mg; J—0.75 mg; Se—0.20 mg. Substances improving digestibility: Endo-1.4.beta-xyl—300 U; Subtilisin—4.000 U; Alpha-amyl—400 U; Endo-1.4-beta-xylanase—1.525 U; Endo-1.3(4)-beta-glucanase—190 U; 6-phytase—500 FTU.

**Table 2 animals-13-00499-t002:** Chemical composition (%) of wheat germ expeller (mean, SD ^1^).

Items	Unit	Wheat Germ Expeller
Gross energy	MJ kg^−1^	14.14
Crude protein	%	35.6 ± 2.5
Crude fat	%	6.0 ± 0.5
Carbohydrates	%	27.8
Crude ash	%	4.75 ± 0.20
Water and volatiles	%	11.6 ± 0.8
Total fiber	%	2.8 ± 0.3

^1^ SD—standard deviation.

**Table 3 animals-13-00499-t003:** Chemical composition of the experimental diets (applies to the finisher’s diet).

Items	Unit	CT ^1^	EX5 ^2^	EX10 ^3^	EX15 ^4^
Gross energy	MJ kg^−1^ DM	19.71	19.80	19.70	19.82
Dry matter	%	90.59	90.66	91.51	91.71
Crude protein	%	23.38	23.46	23.14	22.98
Ether extract	%	6.06	6.09	6.67	7.15
Crude ash	%	5.31	4.84	5.91	5.72
Nitrogen free extractives	%	52.77	53.13	52.58	53.28
Crude fiber	%	3.07	3.14	3.21	2.63
Crude Protein: Energy ratio		1.19	1.18	1.17	1.16
	mg kg^−1^ DM	Macroelements			
Pabs (absorbable phosphorus)		5980.0	609.00	7010.0	6970.0
Na		1199.4	1161.2	1170.2	1175.5
Ca		7582.3	8537.6	10385.5	9086.2
K		488.0	477.0	435.0	447.0
Mg		161.0	181.0	174.0	177.0
			Microelements		
Fe		173.4	165.1	180.7	184.3
Zn		74.2	73.6	80.5	78.1
Cu		14.4	11.3	11.8	11.8
Mn		78.9	86.4	90.1	94.9

^1^ CT, Control Treatment; ^2^ EX5, Experimental diet with 5% wheat germ expeller; ^3^ EX10, Experimental diet with 10% wheat germ expeller; ^4^ EX15, Experimental diet with 15% wheat germ expeller.

**Table 4 animals-13-00499-t004:** Feed intake and weight gain of male 43-day-old Ross-308 broilers (MEAN, SEM, *n* = 32).

Items	CT ^1^	EX5 ^2^	EX10 ^3^	EX15 ^4^	SEM ^5^	*p* Value ^6^
Final body live weight (g)	2514.4 ^a^	2379.4	2270.6 ^b^	2264.4 ^b^	35.8	0.042
Feed consumption (g per 100 g body weight^−1^ )	169.7 ^B^	194.1 ^A^	197.1 ^A^	193.9 ^A^	0.004	<0.001
Body weight gain (g per 100 g feed^−1^ )	58.2 ^A^	51.2 ^B^	50.1 ^B^	50.8 ^B^	0.854	<0.001
Gross energy intake (kJ per 100 g body weight^−1^ )	32.5 ^B,b^	36.3 ^B,a^	41.2 ^A^	34.7 ^B^	0.720	<0.001
Protein intake (g per 100 g body weight^−1^ )	39.7 ^B,b^	45.5 ^A^	45.6 ^A^	44.6 ^a^	0.686	<0.001
Fat intake (g per 100 g body weight^−1^ )	10.3 ^B^	11.8 ^A,D,b^	13.1 ^A,a^	13.9 ^A,C^	0.284	<0.001
Protein efficiency ratio (PER)	2.49 ^A,a^	2.18 ^B^	2.16 ^B^	2.21 ^b^	0.036	<0.001
Feed conversion efficiency (FCE)	0.58 ^A^	0.51 ^B^	0.50 ^B^	0.51 ^B^	0.008	<0.001
Feed conversion ratio (FCR)	1.70 ^B^	1.94 ^A^	1.97 ^A^	1.94 ^A^	0.030	<0.001
Macroelements consumption (mg per 100 g body weight^−1^ )						
P	1014.7 ^B,D^	1181.9 ^B,C^	1381.9 ^A^	1351.4 ^A^	30.1	<0.001
Na	2035.2 ^B,b^	2253.6 ^a^	2306.9 ^A^	2279.1 ^a^	32.7	0.005
Ca	12,899.0 ^B,D^	16,568.9 ^B,C^	20473.9 ^A^	17616.8 ^A^	529.4	<0.001
K	828.1 ^b^	925.7 ^a^	857.6	866.7	12.3	0.038
Mg	273.2 ^B^	351.3 ^A^	343.0 ^A^	343.2 ^A^	6.95	<0.001
Microelements consumption (mg per 100 g body weight^−1^ )						
Fe	294.2 ^B^	320.4 ^b^	356.2 ^A,a^	357.4 ^A,a^	8.13	<0.001
Zn	125.8 ^B^	142.8 ^A,b^	158.8 ^A,a^	151.3 ^A^	2.79	<0.001
Cu	24.3	23.8	23.2	22.8	0.29	0.316
Mn	133.8 ^B^	168.1 ^A^	177.6 ^A^	183.9 ^A^	3.99	<0.001

^1^ CT, Control Treatment; ^2^ EX5, Experimental diet with 5% wheat germ expeller; ^3^ EX10, Experimental diet with 10% wheat germ expeller; ^4^ EX15, Experimental diet with 15% wheat germ expeller; ^5^ SEM, standard error of the mean; BW: body weight; ^6^ Means within a row followed by different superscript letters differ significantly ^A,B,C,D^
*p* ≤ 0.01; ^a,b^
*p* ≤ 0.05.

**Table 5 animals-13-00499-t005:** Chemical composition of breast and thigh of male 43-day-old Ross-308 broilers (MEAN, SEM, *n* = 32).

Items	Meat	CT ^1^	EX5 ^2^	EX10 ^3^	EX15 ^4^	Total	SEM ^5^	Level of Significance		
								Meat (M)	Group (G)	M × G
Gross energy (MJ/100 g)	breast	2.40	2.42	2.40	2.42	2.41 ^Y^	0.01 0.02 0.02	<0.001	0.240	0.405
	thigh	2.68	2.69	2.71	2.71	2.70 ^X^				
	Total	2.54	2.55	2.56	2.57	2.55				
	SEM	0.04	0.04	0.04	0.04					
Moisture (% DM)	breast	74.4	73.1	73.1	73.3	73.5 ^X^	0.20 0.31 0.21	<0.001	<0.001	0.118
	thigh	73.4 ^A^	71.9	70.6 ^B^	70.8 ^B^	71.7 ^Y^				
	Total	73.9 ^A,a^	72.5 ^b^	71.8 ^B^	72.0 ^B^	72.6				
	SEM	0.44	0.19	0.38	0.45					
Crude protein (% DM)	breast	81.6	80.8	82.1	81.5	81.5 ^X^ 64.0 ^Y^ 72.8	0.37 0.45 1.14	<0.001	0.037	0.049
	thigh	66.0 ^A^	64.0	64.0	62.0 ^B^					
	Total	73.8 ^a^	72.4	73.1	71.7 ^b^					
	SEM	2.10	2.22	2.36	2.59					
Crude fat (% DM)	breast	8.73	9.76	8.48	9.46	9.11 ^Y^ 27.8 ^X^ 18.4	0.34 1.04 1.21	<0.001	0.217	0.424
	thigh	26.1	27.3	28.6	29.1					
	Total	17.4	15.5	18.5	19.3					
	SEM	2.31	2.32	2.62	2.60					
Crude ash (% DM)	breast	3.73	3.73	3.51	3.44	3.60 ^Y^ 4.52 ^X^ 4.06	0.05 0.08 0.07	<0.001	0.674	0.002
	thigh	4.23 ^b^	4.37	4.78 ^a^	4.69					
	Total	3.97	4.05	4.15	4.06					
	SEM	0.08	0.11	0.13	0.20					

^1^ CT, Control Treatment; ^2^ EX5, Experimental diet with 5% wheat germ expeller; ^3^ EX10, Experimental diet with 10% wheat germ expeller; ^4^ EX15, Experimental diet with 15% wheat germ expeller; ^5^ SEM, standard error of the mean; Means within a row followed by different superscript letters differ significantly; ^A,B^
*p* ≤ 0.01; ^a,b^
*p* ≤ 0.05; Means within a column followed by different superscript letters differ significantly ^X,Y^
*p* ≤ 0.01; *p* ≤ 0.05.

**Table 6 animals-13-00499-t006:** Macroelements composition (mg/100 g DM) of breast and thigh of male 43-day-old Ross-308 broiler (MEAN, SEM, *n* = 32).

Item	Meat	CT ^1^	EX5 ^2^	EX10 ^3^	EX15 ^4^	Total	SEM ^5^	Level of Significance		
								Meat (M)	Group (G)	M × G
P	breast	827.3	875.3	873.5	844.8	855.2 ^X^ 714.8 ^Y^ 785.0	8.9 9.1 19.9	<0.001	0.007	0.016
	thigh	682.3 ^B^	686.3 ^b^	732.5	758.2 ^A,a^					
	Total	754.8 ^b^	780.8 ^a^	803.0 ^a^	801.5					
	SEM	21.6	25.8	20.2	18.4					
Na	breast	292.7 ^B,b^	380.8 ^a^	400.5 ^A^	384.1 ^a^	364.5 ^X^ 310.7 ^Y^ 337.6	12.6 8.20 8.20	<0.001	0.004	<0.001
	thigh	335.6 ^A^	365.9 ^A^	270.7 ^B^	270.7 ^B^					
	Total	314.1 ^B^	337.4 ^A,a^	335.6	327.4 ^b^					
	SEM	13.0	15.0	18.5	16.3					
Ca	breast	19.8 ^b^	26.2 ^A^	22.6 ^A,a^	15.5 ^B^	21.0 ^Y^ 37.3 ^X^ 29.1	1.01 1.24 1.29	<0.001	0.001	0.001
	thigh	32.0	37.4	39.7	38.8					
	Total	25.9 ^B^	31.8 ^A^	31.1	27.6 ^B^					
	SEM	2.10	1.86	2.82	3.26					
K	breast	848.2	644.8	694.9	632.0	705.0 ^X^ 458.3 ^Y^ 581.6	27.1 19.2 22.6	<0.001	<0.001	0.155
	thigh	600.4 ^A^	425.3 ^B^	384.8 ^B^	422.6 ^B^					
	Total	724.3 ^A^	535.0 ^B^	539.9 ^B^	527.3 ^B^					
	SEM	47.3	40.7	42.1	33.7					
Mg	breast	95.8 ^B,b^	105.0	109.7 ^A^	105.6 ^a^	104.0 ^X^ 86.2 ^Y^ 95.1	2.70 3.08 2.21	<0.001	0.062	0.474
	thigh	84.4	87.2	89.6	83.7					
	Total	90.1 ^b^	96.1	99.7 ^a^	94.6					
	SEM	2.41	2.84	3.58	3.78					

^1^ CT, Control Treatment; ^2^ EX5, Experimental diet with 5% wheat germ expeller; ^3^ EX10, Experimental diet with 10% wheat germ expeller; ^4^ EX15, Experimental diet with 15% wheat germ expeller; ^5^ SEM, standard error of the mean; Means within a row followed by different superscript letters differ significantly; ^A,B,^
*p* ≤ 0.01; ^a,b^
*p* ≤ 0.05; Means within a column followed by different superscript letters differ significantly ^X,Y^
*p* ≤ 0.01; *p* ≤ 0.05.

**Table 7 animals-13-00499-t007:** Microelements composition (mg/100 g DM) of breast and thigh of male 43-day-old Ross-308 broiler (MEAN, SEM, *n* = 32).

Item	Meat	CT ^1^	EX5 ^2^	EX10 ^3^	EX15 ^4^	Total	SEM ^5^	Level of Significance		
								Meat (M)	Group (G)	M × G
Fe	breast	8.51	8.94	7.57	7.67	8.17 ^X^ 6.91 ^Y^ 7.54	0.28 0.27 0.21	<0.001	0.075	0.047
	thigh	7.36	6.15	6.11 ^b^	8.02 ^a^					
	Total	7.93	7.54	6.84	7.84					
	SEM	0.25	0.59	0.42	0.32					
Zn	breast	2.81 ^a^	2.42	2.34 ^b^	2.59	2.54 ^Y^ 4.53 ^X^ 3.53	0.07 0.09 0.14	<0.001	<0.001	0.216
	thigh	5.09 ^A^	4.56 ^A^	3.91 ^B^	4.56 ^A^					
	Total	3.95 ^A,a^	3.49 ^b^	3.13 ^B,c^	3.57 ^A,b^					
	SEM	0.31	0.28	0.22	0.27					
Cu	breast	1.11	1.23	1.00	0.91	1.06 ^X^ 0.70 ^Y^ 0.88	0.05 0.05 0.04	<0.001	0.384	0.122
	thigh	0.80	0.61	0.68	0.72					
	Total	0.96	0.92	0.84	0.81					
	SEM	0.05	0.12	0.08	0.05					
Mn	breast	0.19	0.15	0.14	0.15	0.15 ^X^ 0.13 ^Y^ 0.14	0.01 0.01 0.01	0.009	0.003	0.544
	thigh	0.15	0.12	0.11	0.14					
	Total	0.17 ^A,a^	0.13 ^b^	0.12 ^B^	0.14					
	SEM	0.01	0.01	0.01	0.01					

^1^ CT, Control Treatment; ^2^ EX5, Experimental diet with 5% wheat germ expeller; ^3^ EX10, Experimental diet with 10% wheat germ expeller; ^4^ EX15, Experimental diet with 15% wheat germ expeller; ^5^ SEM, standard error of the mean; Means within a row followed by different superscript letters differ significantly; ^A,B^
*p* ≤ 0.01; ^a,b,c^
*p* ≤ 0.05; Means within a column followed by different superscript letters differ significantly ^X,Y^
*p* ≤ 0.01; *p* ≤ 0.05.

**Table 8 animals-13-00499-t008:** Nutrient reference values-requirements (NRVs-R mg per 100 g) and their implementation (%) in the breast and thigh of a male 43-day-old Ross-308 broiler (mean, SEM, *n* = 32).

Item	Meat	NRVs-R^1^	CT ^2^	EX5 ^3^	EX10 ^4^	EX15 ^5^	Total	SEM ^6^	Level of Significance		
									Meat (M)	Group (G)	M × G
P	breast	700	118.2	125.0	124.8	120.7	122.2 ^X^ 102.1 ^Y^ 112.1	1.27 1.30 1.55	<0.001	0.007	0.016
	thigh		97.5 ^B^	98.0 ^b^	104.6	108.3 ^A,a^					
	Total		107.8 ^b^	111.5	114.7 ^a^	114.5 ^a^					
	SEM		3.09	3.68	2.89	2.63					
Ca	breast	1000	1.98 ^b^	2.62 ^A,a^	2.26 ^A^	1.55 ^B^	2.10 ^Y^ 3.72 ^X^ 2.91	0.10 0.12 0.13	<0.001	0.002	<0.001
	thigh		3.20	3.74	3.97	3.98					
	Total		2.59 ^B^	3.18 ^A,a^	3.11	2.76 ^b^					
	SEM		0.21	0.19	0.28	0.33					
Mg	breast	310	30.9 ^B^	33.9	35.4 ^A^	34.0 ^A^	33.6 ^X^ 27.8 ^Y^ 30.7	0.48 0.59 0.52	<0.001	0.061	0.472
	thigh		27.2	28.1	28.9	27.0					
	Total		29.1 ^b^	31.0	32.2 ^a^	30.5					
	SEM		0.78	0.92	1.16	1.22					
Fe	breast	14	60.8	63.9	54.0	54.8	58.4 ^X^ 49.3 ^Y^ 53.9	2.00 1.95 1.50	<0.001	0.069	0.049
	thigh		52.6	43.9	43.6 ^b^	57.3 ^a^					
	Total		56.7	53.9	48.8	56.0					
	SEM		1.79	4.23	2.98	2.26					
Zn	breast	11	25.6 ^a^	22.0	21.3 ^b^	23.5	23.1 ^Y^ 41.2 ^X^ 32.1	0.59 0.85 1.25	<0.001	<0.001	0.303
	thigh		46.2 ^A^	41.5 ^A^	35.5 ^B^	41.5 ^A^					
	Total		35.9 ^A^	31.5 ^B^	28.4 ^B,D^	32.5 ^C^					
	SEM		2.81	2.55	2.03	2.42					
Cu	breast	0.9	123.0	136.4	110.0	100.6	117.7 ^X^ 78.3 ^Y^ 98.0	5.27 5.10 4.40	<0.001	0.315	0.113
	thigh		89.3	67.8	75.8	80.4					
	Total		106.2	102.1	93.4	90.5					
	SEM		5.92	12.8	9.32	5.45					
Mn	breast	3	6.17	4.84	4.52	4.87	5.10 ^X^ 4.29 ^Y^ 4.70	0.23 0.23 0.17	0.006	0.003	0.416
	thigh		4.97	3.88	3.53	4.79					
	Total		5.57 ^A,a^	4.36 ^B^	4.03 ^b^	4.83					
	SEM		0.27	0.36	0.29	0.32					

^1^ NRVs-R, Nutrient Reference Values-Requirements; ^2^ CT, Control Treatment; ^3^ EX5, Experimental diet with 5% wheat germ expeller; ^4^ EX10, Experimental diet with 10% wheat germ expeller; ^5^ EX15, Experimental diet with 15% wheat germ expeller; ^6^ SEM, standard error of the mean; Means within a row followed by different superscript letters differ significantly; ^A,B,C,D^
*p* ≤ 0.01; ^a,b^
*p* ≤ 0.05; Means within a column followed by different superscript letters differ significantly ^X,Y^
*p* ≤ 0.01; *p* ≤ 0.05.

## Data Availability

The data presented in this study are available on request from the corresponding author.

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
