# Peer review of "The Energy and Nutritional Value of Meat of Broiler Chickens Fed with Various Addition of Wheat Germ Expeller"

_animals, 2023, doi:10.3390/ani13030499_

Round 1

Reviewer 1 Report

Although the study is exploring a very novel feed additive which must be investigated but  

1.      1. Sample size for different analysis (08 birds) per group is very small, as mentioned by the authors it is a pilot study, so a larger experiment must be conducted before propagating the results of the study.

2.      2. The abstract is not mentioning results pertaining to growth performance, unless the reader goes to the result and discussion section.

3.      3. As this article has been submitted to a special issue "Role of Trace Element in Animal Health and Metabolic". The article, at least for me, is not related to the theme of the special issue which, according to my understanding, is animal health related studies pertaining to the role of trace elements. So, by only analyzing the mineral composition of broiler meat, this study does not fit in the theme of special issue.

4.      4. The authors has not shown uniformity in writing abbreviations in the manuscript, authors used many abbreviations directly without giving full form at first use in text. E.g line 46.

5.      5. Need improvement regarding English grammar.

Reviewer 2 Report

The consumption of broiler meat is increasing worldwide and in tendency to increase in the future. Feed industry and the researchers are trying to find new protein sources. This study investigated the use wheat germ expeller as an alternative feed additive in broiler feeding. the hypothesis was well described and paper was good written. It will be useful for researchers and readers looking for alternative sources/additives in poultry feeding.

Reviewer 3 Report

The work takes into consideration too many parameters and the objectives of the work is not clear.

It is not clear what the consumer benefit of chicken meat is.

It does not list which vaccines were given when the chickens were allocated

x

Round 2

Reviewer 3 Report

is accepted in this form with the additions that have be made  

Author Response

Thank you for your comment.